# Species-Specific Stress Responses to Selenium Nanoparticles in *Pseudomonas aeruginosa* and *Proteus mirabilis*

**DOI:** 10.3390/nano15181404

**Published:** 2025-09-12

**Authors:** Kidon Sung, Miseon Park, Ohgew Kweon, Alena Savenka, Angel Paredes, Monica Sadaka, Saeed Khan, Seonggi Min, Steven Foley

**Affiliations:** 1Division of Microbiology, National Center for Toxicological Research, U.S. Food and Drug Administration, Jefferson, AR 72079, USA; miseon.park@fda.hhs.gov (M.P.); oh-gew.kweon@fda.hhs.gov (O.K.); saeed.khan@fda.hhs.gov (S.K.); steven.foley@fda.hhs.gov (S.F.); 2Nanotechnology Core Facility, Office of Scientific Coordination, National Center for Toxicological Research, U.S. Food and Drug Administration, Jefferson, AR 72079, USA; alena.savenka@fda.hhs.gov (A.S.); angel.paredes@fda.hhs.gov (A.P.); 3University of Arkansas, Little Rock, AR 72204, USA; mssadaka@ualr.edu; 4Division of Biochemical Toxicology, National Center for Toxicological Research, U.S. Food and Drug Administration, Jefferson, AR 72079, USA; seonggi.min@fda.hhs.gov

**Keywords:** Selenium, nanoparticles (NPs), antibacterial, urinary pathogens, proteome

## Abstract

Urinary tract infections (UTIs) remain a major global health concern, with rising antimicrobial resistance prompting the search for alternative therapies. Selenium nanoparticles (Se NPs) are promising antimicrobial agents due to their unique physicochemical properties and ability to disrupt bacterial physiology. This study evaluated the antibacterial efficacy of Se NPs against four uropathogens and conducted comparative proteomic analyses to elucidate stress responses. Enumeration assays showed that Se NPs effectively inhibited bacterial growth, with *Pseudomonas aeruginosa* being the most susceptible and *Proteus mirabilis* the most resistant. Microscopy revealed Se NP-induced membrane rupture and cellular deformation across all species. Proteomic and bioinformatic analyses showed more pronounced protein regulation in *P. mirabilis* than in *P. aeruginosa*. Cluster of Orthologous Groups (COG) analysis revealed both shared and species-specific responses, while Kyoto Encyclopedia of Genes and Genomes (KEGG) pathway analysis indicated activation of key stress pathways. Virulence-associated proteins were modulated in both species, with *P. mirabilis* uniquely upregulating stress survival and exotoxin-related proteins. Both regulated efflux pumps, suggesting active transport mitigates Se NP toxicity. *P. aeruginosa* showed mercury resistance, while *P. mirabilis* expressed tellurite resistance proteins. These findings highlight distinct yet overlapping strategies and support the potential of Se NPs in novel antimicrobial development.

## 1. Introduction

Urinary tract infections (UTIs) are a pervasive global health issue, affecting approximately 400 million individuals annually [1]. In the United States alone, UTIs impose an estimated $3.5 billion in healthcare costs each year [2]. These infections encompass a range of conditions, including urethritis, cystitis, prostatitis, ureteritis, and pyelonephritis, which can progress to severe complications such as bacteremia and septicemia [3]. UTIs account for 1–6% of all healthcare visits annually, including millions of medical consultations. Women are disproportionately affected, with at least 50% experiencing a UTI during their lifetime [4]. Additionally, UTIs are associated with significant morbidity, particularly among vulnerable populations such as the elderly. This widespread prevalence and associated economic burden highlight the urgent need for improved prevention, management, and treatment strategies to reduce the impact of UTIs on individuals and public health overall.

UTIs are commonly caused by a variety of pathogens, including Escherichia coli, Klebsiella pneumoniae, Proteus mirabilis, Citrobacter, Pseudomonas aeruginosa, Staphylococcus aureus, Enterococcus faecalis, Streptococcus bovis, and Candida albicans [5]. The widespread use of antibiotics to treat these infections has driven the alarming emergence of antibiotic resistance [6], with multidrug-resistant (MDR) *E. coli* posing a significant clinical challenge that often requires reliance on broad-spectrum antibiotics. Reliance on broad-spectrum treatments not only increases the risk of adverse effects but also contributes to prolonged hospital stays and higher treatment costs. The escalating threat of antibiotic resistance underscores the urgent need to develop alternative antimicrobial strategies for the treatment and prevention of UTIs. Developing and implementing these alternatives may offer viable solutions to reduce antibiotic dependence and combat the rise in resistant uropathogens.

Nanotechnology has emerged as a transformative approach in biomedicine, offering innovative solutions to address antimicrobial resistance [7]. Among emerging advancements, nanoparticle (NP)-based antibacterial strategies have gained prominence as a novel approach, offering distinct mechanisms of action that lower the likelihood of bacterial resistance compared to conventional antibiotics. Beyond their broad-spectrum antimicrobial activity, NPs exhibit unique physicochemical properties, including high surface area-to-volume ratios and tunable surface functionalities, which enhance their efficacy in biomedical applications [8]. These attributes enable NPs to disrupt bacterial membranes, induce reactive oxygen species generation, and facilitate targeted antimicrobial delivery, thereby improving therapeutic outcomes.

The integration of nanotechnology into antimicrobial strategies presents a promising avenue for overcoming MDR bacterial infections, providing an effective alternative to traditional antibiotics and expanding the arsenal against resistant pathogens. Selenium (Se) is an essential micronutrient with potent antioxidant properties that protect cellular membranes by neutralizing harmful free radicals [9]. Se plays a crucial role in immune system function, contributing to lymphocyte activation, proliferation, and differentiation [10]. Se NPs are gaining significant interest due to their exceptional biocompatibility and low toxicity profiles [11].

Recent studies have demonstrated that Se NPs possess broad-spectrum antimicrobial activity. Filipović et al. reported that Se NPs with different stabilizers showed size- and surface-dependent antibacterial effects against *E. coli*, *P. aeruginosa*, *S. aureus*, and *E. faecalis* [12]. Han et al. demonstrated that Se NPs and selenium nanowires exhibit potent activity against multidrug-resistant strains such as MRSA and VRE, with enhanced effects when combined with antibiotics like linezolid [13]. Ridha et al. further confirmed that Se NPs inhibit both planktonic growth and biofilm formation of *S. aureus*, *S. epidermidis*, and *P. aeruginosa* [14]. More recently, Salah et al. showed that polyvinylpyrrolidone (PVP)-stabilized Se NPs not only possess strong antibacterial activity at low MIC values but also induce structural damage visible by microscopy [15]. Collectively, these studies underscore the antimicrobial potential of Se NPs but also reveal gaps in understanding the species-specific stress responses and adaptive mechanisms triggered by Se NP exposure. To place these findings in the broader context of nanotechnology-enabled UTI therapeutics, a summary of recent nanoparticle-based strategies is provided in Table 1.

Building on these findings, the present study evaluates the antibacterial efficacy of Se NPs against four clinically significant uropathogens—*E. coli*, *E. faecalis*, *P. mirabilis*, and *P. aeruginosa*—while employing global quantitative proteomic analysis to elucidate molecular mechanisms underlying bacterial responses. By integrating bacterial enumeration assays, microscopy, and proteomics, this work provides new mechanistic insights into species-specific stress adaptations, resistance determinants, and virulence regulation under Se NP-induced stress. These results advance our understanding of nanoparticle–pathogen interactions and highlight the potential of Se NPs as alternative therapeutics against multidrug-resistant uropathogens.

## 2. Materials and Methods

### 2.1. Antibacterial Activity

Se beads (99.99%, <5 mm diameter, MilliporeSigma, Burlington, MA, USA) were used as targets for laser ablation synthesis in deionized (DI) water (~8 mm depth) using a Nd:YAG laser (Electro Scientific Industries, Portland, OR, USA) at a 45° angle (Appendix A). Laser parameters were set at repetition rates of 1–15 kHz with pulse energies of 16.5 mJ at 1 kHz. Se NPs was generously provided by Professor Gregory Guisbiers, School of Physical Sciences, University of Arkansas at Little Rock, AR. Detailed synthesis and characterization of Se NPs have been previously reported [31].

The bacterial strains used in this study included *Escherichia coli* (ATCC 700928), *Proteus mirabilis* (ATCC 7002), and *Enterococcus faecalis* (ATCC 29212), all purchased from the American Type Culture Collection (ATCC, Manassas, VA, USA). Additionally, *Pseudomonas aeruginosa* PA14 was generously provided by Professor Vincent Lee of the Department of Cell Biology and Molecular Genetics at the University of Maryland, College Park. Overnight cultures of these strains were prepared by inoculating single colonies into tryptic soy broth (TSB, Thermo Fisher Scientific, Waltham, MA, USA) and incubating at 37 °C with shaking. The optical density at 600 nm was adjusted to 0.01 before treating the bacterial suspension with 32 ppm Se NPs for 10, 30, and 60 min. A positive control (testing medium with bacteria) and a negative control (medium only) were included. Following treatment, 100 µL aliquots were serially diluted, plated on tryptic soy agar (TSA, Thermo Fisher Scientific), and incubated overnight at 37 °C. The number of viable bacterial colony-forming units (CFUs) was then quantified. All experiments were performed in triplicate. Appendix A illustrates the methodology used to assess antibacterial activity. Statistical significance of bacterial CFU reductions over time was assessed using one-way analysis of variance followed by post hoc pairwise *t*-tests.

### 2.2. Field Emission Scanning Electron Microscopy (Fesem)

Bacterial suspensions were treated with 32 ppm Se NPs under the same conditions described in subchapter 2.1 (60 min of exposure). Following Se NP treatment, bacterial suspensions were centrifuged at 10,000 rpm for one minute, and the precipitates were washed three times with phosphate-buffered saline (PBS, Thermo Fisher Scientific). The samples were then fixed in 5 mL of 3% (*v*/*v*) glutaraldehyde (Electron Microscopy Sciences, Hatfield, PA, USA) prepared in 0.1 M sodium cacodylate buffer (pH 7.2, Electron Microscopy Sciences), and incubated for 24 h. Following primary fixation, samples were centrifuged at 8000 rpm for 2 min, and the fixative was carefully removed. The fixed bacterial pellets were washed three times with 0.1 M sodium cacodylate buffer (pH 7.2) for 10 min each wash to remove excess glutaraldehyde. Between each wash, samples were centrifuged at 8000 rpm for 2 min. Dehydration was performed using seven ethanol gradients (15%, 30%, 50%, 70%, 90%, 95%, 100%), with each step lasting 15 min. The 100% ethanol step was repeated twice to ensure complete water removal. After the removal of 100% ethanol, samples underwent further dehydration through sequential immersion in hexamethyldisilazane (HMDS, Electron Microscopy Sciences) and ethanol (100%) solutions. First, samples were incubated in a 1:1 solution of HMDS and ethanol for 15 min, followed by a 2:1 HMDS/ethanol solution for another 15 min. Next, the samples were immersed in HMDS for 20 min, a step that was repeated to ensure complete dehydration. Finally, the samples were left in HMDS and allowed to air dry overnight in a fume hood. The gradual evaporation of HMDS preserves the three-dimensional structure of bacterial cells without the surface tension effects that typically occur during air drying from aqueous solutions. Once completely dried, the bacterial samples were mounted onto specimen stubs using carbon adhesive discss. The samples were evenly distributed across the stub surface to ensure optimal imaging area. Care was taken to avoid sample contamination and to minimize electrostatic buildup during mounting. A thin conductive coating of gold–palladium alloy was then applied to the fixed samples using a sputter coater (Denton Vacuum, Moorestown, NJ, USA) to enhance conductivity for imaging. The coated samples were then examined in high vacuum mode using a Zeiss-Merlin Gemini2 FESEM (Carl Zeiss Microscopy, White Plains, NY, USA). Images were acquired at an accelerating voltage of 5 kV with a working distance of 3–5 mm using a secondary electron (SE) detector to achieve high-resolution surface topography. Multiple magnifications (5000× to 50,000×) were employed to visualize bacterial surface morphology, cell wall integrity, and potential structural alterations such as membrane disruption, pore formation, or cell lysis following Se NP treatment.

### 2.3. Protein Sample Preparation

Based on the antibacterial test results, proteomic analysis was performed on *P. aeruginosa* PA14, which exhibited the highest susceptibility to Se NPs, and *P. mirabilis* ATCC 7002, which showed the lowest susceptibility. Following Se NP treatment, bacterial cells were centrifuged at 14,000 rpm for one minute at 4 °C, washed with PBS, and lysed using the BugBuster Plus Lysonase kit (MilliporeSigma) in Lysing Matrix tubes via FastPrep-24 homogenization. The lysate was further disrupted by boiling and vortexing, followed by centrifugation at 14,000 rpm for 30 min to obtain the protein extract. Proteins were precipitated with trichloroacetic acid, solubilized in 8 M urea and 50 mM Tris-HCl (pH 8.0), and supplemented with protease inhibitors.

Reduction and alkylation were performed using dithiothreitol and iodoacetamide, respectively. Proteins were digested with trypsin at a 1:20 enzyme-to-substrate ratio, acidified with 0.3% trifluoroacetic acid (TFA), and subjected to solid-phase extraction (SPE) using μHLB cartridges. The resulting eluates were frozen, lyophilized, and reconstituted in 0.1% TFA for subsequent analysis. All chemicals required for protein sample preparation were obtained from MilliporeSigma.

### 2.4. Liquid Chromatography with Tandem Mass Spectrometry (Lc-Ms/MS) and Bioinformatic Analysis

One microgram of the peptide pool was analyzed by nano-liquid chromatography–mass spectrometry (LC-MS) using a Waters M-class high-performance liquid chromatography (HPLC) system coupled to an Orbitrap Exploris 480 Mass Spectrometer (Thermo Fisher Scientific). Peptides were separated on a 75 μm C18 analytical column at 55 °C and eluted with a 30 min gradient at 350 nL/min. The mass spectrometer operated in data-independent acquisition (DIA) mode, collecting six gas-phase fractionation (GPF) injections across defined mass ranges. Each GPF included a full MS scan (60,000 resolution) and 26 MS/MS scans with staggered isolation windows (30,000 resolution). MS parameters included an automated gain control (AGC) target of 1e6, a maximum ion injection time of 50 ms, and NCE set to 30. For sample analysis, the same LC-MS conditions were used, but with a DIA acquisition method featuring full MS scans (*m*/*z* 385–1015) at 60,000 resolution, followed by 61 MS/MS scans at 15,000 resolution. DIA data were processed using Scaffold DIA (Proteome Software, Portland, OR, USA), with RAW files converted to mzML format and aligned for retention time consistency. Peptide identification used the Prosit library, applying a 1% false discovery rate via Percolator [32].

Quantitation was performed using EncyclopeDIA [33], selecting the top five fragment ions. Protein expression was considered significant if the fold change was ≥2.0 (upregulated) or ≤0.5 (downregulated). Functional annotation was conducted using the Cluster of Orthologous Groups (COG) database, while Kyoto Encyclopedia of Genes and Genomes (KEGG) pathway analysis linked differentially expressed proteins to biological processes [34,35]. A custom Python script was employed to cluster protein expression (EPN) data across treatment time points [36].

## 3. Results and Discussion

### 3.1. Antibacterial Activity of SeNPs

The antibacterial effects of Se NPs varied significantly among the tested uropathogens, revealing species-specific differences in susceptibility. Bacterial enumeration assays revealed that Se NPs exerted a notable inhibitory effect on bacterial CFU counts, but the magnitude of inhibition differed among species (Figure 1). *E. faecalis* showed moderate reductions at 30 min (*p* < 0.05) and stronger inhibition at 60 min (*p* < 0.01). *P. mirabilis* was more resilient, with no significant effect at 10 min but significant decreases at 30 min (*p* < 0.01) and 60 min (*p* < 0.001). *E. coli* exhibited progressive inhibition, significant at 30 min (*p* < 0.05) and 60 min (*p* < 0.01). *P. aeruginosa* was the most susceptible, showing highly significant reductions at 10, 30, and 60 min (*p* < 0.001 for all). In all cases, positive control samples showed significantly higher CFU counts than Se NP-treated samples.

To further assess the impact of Se NP exposure on bacterial morphology, we performed FESEM to visualize structural changes at the cellular level. Imaging of untreated bacterial cells revealed smooth surfaces and intact cellular structures, indicative of healthy bacterial morphology (Figure 2A,C,E,G). However, after 60 min of Se NP exposure, all four bacterial species exhibited varying degrees of structural damage, suggesting significant morphological alterations in response to NP-induced stress (Figure 2B,D,F,H). Such morphological disruption points to bacterial membranes as a key target of Se NP action, leading to loss of integrity, leakage of intracellular contents, and eventual cell death.

These findings are consistent with prior reports that have demonstrated the broad-spectrum antibacterial potential of Se NPs. Shakibaie et al. showed that biogenic Se NPs inhibited *P. aeruginosa* and *S. aureus* growth, attributing their effects to oxidative stress and membrane damage [37]. Xu et al. reported that Se NPs penetrated bacterial membranes, triggering ROS-mediated toxicity [38]. More recently, Lin et al. confirmed Se NP-induced ROS generation and membrane disruption in *S. aureus* [39].

Taken together, these results reinforce the antibacterial potential of SeNPs while emphasizing that their efficacy is pathogen-dependent. The morphological and quantitative evidence provided here adds to the growing body of literature supporting SeNPs as promising antimicrobial candidates.

### 3.2. Comparative Proteome Profile

To gain a deeper understanding of bacterial adaptive mechanisms in response to Se NP exposure, we conducted a global proteomic analysis on *P. aeruginosa* and *P. mirabilis*, which exhibited the highest and lowest susceptibility, respectively. The proteomic profiling revealed distinct yet overlapping patterns of protein expression, demonstrating species-specific metabolic adjustments under NP-induced stress. Proteins showing no significant change in expression in *P. aeruginosa* ranged from 35.15% (2081 proteins) at 60 min to 37.50% (2220 proteins) at 30 min, while in *P. mirabilis*, these ranged from 33.21% (1206 proteins) at 30 min to 38.64% (1403 proteins) at 10 min (Figure 3A,B). The number of upregulated proteins was substantially higher in *P. mirabilis*, peaking at 30 min with 11.02% (400 proteins), compared to *P. aeruginosa*, which showed only 3.87% (229 proteins) at 60 min. This discrepancy suggests that *P. mirabilis* undergoes a more dynamic and rapid metabolic shift in response to Se NP-induced stress, likely reflecting an enhanced ability to modulate its metabolic pathways to counteract environmental stressors.

### 3.3. COG Functional Analysis

COG functional analysis of differentially expressed proteins revealed both shared and unique adaptive responses in *P. aeruginosa* and *P. mirabilis* following Se NP treatment (Figure 4A,B). Both *P. aeruginosa* and *P. mirabilis* exhibited significant alterations in protein expression related to core cellular functions, including energy production and conversion, transcriptional regulation, amino acid metabolism, and protein synthesis. These shared metabolic responses suggest that both species employ conserved strategies to maintain cellular homeostasis under NP-induced stress. Enhanced expression of proteins involved in energy metabolism and transcriptional regulation indicates that both bacteria activate stress response pathways that promote adenosine triphosphate (ATP) production and transcriptional reprogramming to sustain viability under adverse conditions.

Despite these similarities, species-specific adaptations were evident. *P. mirabilis* uniquely upregulated proteins associated with cell wall and membrane biogenesis. The enhanced expression of proteins involved in cell envelope synthesis suggests that *P. mirabilis* employs a protective strategy that strengthens its outer membrane and reduces NP permeability, thereby minimizing cellular disruption [40].

A comparison with prior research revealed both conserved and species-specific responses to NP-induced stress. For example, *P. aeruginosa* exposed to copper stress exhibited significant changes in inorganic ion transport and metabolism (P), energy production and conversion (C), amino acid transport and metabolism (E), cell wall/membrane/envelope biogenesis (M), and translation, ribosomal structure, and biogenesis (J) [41]. Similarly, when *Streptococcus suis* was exposed to silver NPs, key adaptive responses included translation, ribosomal structure, and biogenesis (J), cell wall/membrane/envelope biogenesis (M), amino acid transport and metabolism (E), transcription (K), and carbohydrate transport and metabolism (G) [42]. Likewise, *Xanthomonas campestris pv. campestris* under copper stress exhibited significant differential regulation of inorganic ion transport and metabolism (P), carbohydrate transport and metabolism (G), amino acid transport and metabolism (E), transcription (K), and replication, recombination, and repair (L) [43]. The observed similarities with bacterial responses to other NPs and heavy metals suggest that certain stress-adaptive mechanisms are conserved across species, while others are tailored to individual bacterial physiology.

### 3.4. KEGG Pathway Analysis

We conducted KEGG pathway analysis to further investigate the molecular mechanisms underlying the bacterial stress response to Se NP exposure. This analysis revealed significant alterations in metabolic pathways across both *P. aeruginosa* and *P. mirabilis* (Figure 5 and Figure 6; Appendix A). Metabolic pathways, including biosynthesis of secondary metabolites, microbial metabolism in diverse environments, and biosynthesis of amino acids, exhibited the most significant changes, with both upregulated and downregulated proteins in both bacteria. The extensive involvement of these pathways suggests that Se NP exposure induces a global metabolic shift in both species, requiring the reallocation of cellular resources to support survival under oxidative and structural stresses.

Beyond these fundamental cellular processes, several key pathways were markedly overexpressed in both species. Specifically, multiple two-component regulatory systems, along with pathways related to bacterial chemotaxis, flagellar assembly, and ATP-binding cassette (ABC) transporters, were significantly upregulated. These pathways are essential for bacterial sensing, mobility, and active transport [44], suggesting that Se NP exposure prompts both species to enhance environmental sensing and resource allocation mechanisms. This pattern underscores the bacteria’s ability to detect and respond to stress stimuli, regulate cellular motility, and manage transmembrane transport processes, all of which are crucial for mitigating NP-induced toxicity.

The two-component system is a highly conserved bacterial signaling mechanism that enables cells to sense environmental changes and rapidly trigger appropriate regulatory responses [45]. In both *P. aeruginosa* and *P. mirabilis*, Se NP treatment resulted in the upregulation of two-component system-associated proteins, including AlgB, AlgR, CcoN, CydA, CpxA, DctA, FlrC, GlnL, GlnR, NarG, PhoB, PhoR, and RegB. These proteins play essential roles in bacterial stress adaptation by modulating gene expression in response to such external stimuli as oxidative stress, metal ion toxicity, and membrane damage [46].

These findings align with previous findings. For example, Filipović et al. found that SeNPs with different surface chemistries induce distinct bacterial stress responses, including membrane-targeted effects, in *E. coli* and *P. aeruginosa* [12]. Moreover, Vahdati and Moghadam reported synergistic antibacterial effects when SeNPs were combined with lysozyme, among mechanisms that include disruption of bacterial signaling and cell envelope integrity [47].

The observed upregulation of these components suggests that Se NP exposure activates a broad stress-responsive signaling network, enabling bacteria to dynamically regulate gene expression and cellular functions to enhance survival. This response is consistent with findings from previous studies on bacterial adaptation to environmental stressors. For example, *Salmonella* Typhimurium exposed to high-intensity ultrasound exhibited significant upregulation of two-component system proteins [48]. Similarly, *P. aeruginosa* treated with antibiotics showed a comparable upregulation of two-component system components, suggesting that both antibiotic- and NP-induced stresses may elicit overlapping bacterial response mechanisms [49]. Additional evidence from studies on *Staphylococcus epidermidis* under tigecycline treatment and *S. mutans* under bacitracin stress further supports the critical role of this pathway in bacterial survival under diverse environmental challenges [50,51].

Chemotaxis and motility are crucial bacterial behaviors that facilitate movement toward favorable environments and away from harmful stressors [52]. In response to Se NP exposure, proteins associated with bacterial chemotaxis, including CheW, CheZ, FliG, FliM, Mcp, MotA, and RbsB, exhibited significant upregulation. Similarly, flagellar assembly proteins such as FlgB, FlgD, FlgF-H, FlgJ, FlgM, FliD, FliF-H, FliK, FliM, and FlrC were also overexpressed. This widespread upregulation suggests that enhanced bacterial motility is a key survival strategy for evading toxic NP interactions.

Previous studies have demonstrated that bacterial motility is a crucial adaptive mechanism under stress conditions. For instance, *P. aeruginosa* exposed to antibiotics exhibited a significant increase in bacterial chemotaxis gene expression, reinforcing the role of motility in bacterial adaptation [49]. Additionally, *Natranaerobius thermophilus* relied on bacterial chemotaxis to survive high-salt conditions, suggesting that motility plays a pivotal role in microbial stress responses across diverse bacterial species [53].

Bacterial cells also upregulated membrane transport mechanisms in response to Se NP-induced stress. ABC transporters, a superfamily of integral membrane proteins, play a vital role in nutrient uptake, efflux of toxic compounds, and overall bacterial homeostasis by utilizing ATP hydrolysis to transport a wide array of substrates [54]. After Se NP treatment, ABC transporter-associated proteins, including FtsX, LptB, MlaB, PotD, and ZnuA, were significantly upregulated, suggesting that bacteria prioritize transmembrane transport processes to mitigate NP-induced stress. Similarly, bacterial secretion system proteins, such as SecB, ShlA, TatA, TatB, VgrG, and YajC, were overexpressed, further supporting the hypothesis that Se NP exposure triggers an increase in transmembrane transport activity.

ABC transporters have been widely implicated in bacterial stress responses. In *Brucella melitensis*, ABC transporters were upregulated under salt stress to maintain osmotic balance, and in *S. epidermidis*, ABC transporter proteins were overexpressed in response to antibiotics, highlighting their importance in antimicrobial resistance mechanisms [50,55]. Furthermore, *S. mutans* exposed to theaflavins showed significant upregulation of membrane transport pathways, supporting the importance of transport mechanisms in bacterial adaptation to environmental challenges [56]. Interestingly, some studies have reported the downregulation of ABC transporters under acid stress, as observed in *Lactiplantibacillus plantarum*, where ABC transporter pathways were suppressed in response to pH changes [57], suggesting that bacterial responses to environmental stressors are context-dependent.

### 3.5. Expression Pattern (EPN) Analysis

Our EPN analysis revealed distinct yet overlapping adaptive responses. EPN clustering was used to categorize proteins based on their differential expression across time points, providing insights into metabolic and regulatory shifts triggered by NP exposure. *P. aeruginosa* exhibited 26 significant EPN clusters, whereas *P. mirabilis* had 23 clusters, indicating a comparable but distinct response to Se NP-induced stress (Appendix A).

In the EPN2 dataset, which included proteins consistently overexpressed throughout the treatment, *P. aeruginosa* showed upregulation of 73 proteins involved primarily in energy production and conversion (C) (Figure 7A, Appendix A). Among these, key components of the electron transport chain and ATP synthesis machinery, including AtpA, Fpr, GlcE, NapA, NuoM, Qor, RnfC, and RnfG, were markedly overexpressed [58]. The upregulation of these proteins suggests that *P. aeruginosa* responded to Se NP stress by increasing ATP production and optimizing oxidative phosphorylation. Additionally, the heightened expression of electron transport chain components, including NuoM and Qor, suggests a greater reliance on oxidative phosphorylation as a primary strategy to mitigate Se NP exposure-induced oxidative stress [59].

In contrast, *P. mirabilis* exhibited overexpression of 84 proteins in the EPN2 dataset, but with a different functional emphasis (Figure 7B, Appendix A). The most significantly upregulated proteins were associated with cell wall/membrane/envelope biogenesis (M) and translation, ribosomal structure, and biogenesis (J). Proteins such as LptD, MipA, RfaB, RfaQ, Slp, TtgC, and YtfM, which are involved in cell envelope biogenesis [60], were significantly induced. This suggests that *P. mirabilis* reinforced its structural integrity in response to Se NP exposure [61]. Additionally, protein expression involved in translation, such as RpmF, RpmH, RpsS, RpsU, and TrmA, also had greatly increased, indicating that *P. mirabilis* prioritized sustaining ribosomal function to maintain protein synthesis under stress.

The EPN3 dataset, which included proteins consistently downregulated throughout the treatment, revealed further distinctions between the two species. In *P. aeruginosa*, 69 proteins were consistently downregulated, with transcription-related proteins (K) exhibiting the most pronounced decreases (Figure 7A, Appendix A). Key transcriptional regulators such as Anr, ColR, Rnk, SpoT, and YebC showed significant downregulation, suggesting that Se NP exposure suppressed specific regulatory pathways governing stress responses. The suppression of transcriptional regulators such as Anr and ColR indicates a shift away from certain metabolic pathways, likely as an energy conservation strategy in response to stress [62].

In *P. mirabilis*, 40 proteins were consistently downregulated; the most affected category was nucleotide transport and metabolism (F) (Figure 7B, Appendix A). Proteins such as Add, ChbG, Cmk, GpmB, KdsA, PneB, and Udk, which are involved in nucleotide biosynthesis and metabolism [63], exhibited significant decreases in expression.

### 3.6. Differentially Expressed Proteins Associated with Virulence Factors

Se NP treatment resulted in both shared and species-specific regulation of virulence-associated proteins. *P. aeruginosa* exhibited 47 differentially expressed virulence-related proteins, while *P. mirabilis* displayed a broader response, with 59 differentially expressed virulence-associated proteins (Table 2 and Table 3). Despite differences in the number of virulence factors affected, both species exhibited overlapping responses in key functional categories, including adherence, biofilm formation, secretion systems, immune modulation, motility, and nutrient acquisition. However, *P. mirabilis* displayed additional regulation of stress survival- and exotoxin-related proteins. Proteins are involved in capsular polysaccharide synthesis, lipid A modifications, and efflux-mediated immune resistance [64]. The broader immune modulation response in *P. mirabilis* suggests that this species may rely more extensively on modifying its cell surface structures to resist NP-induced stress. Previous research has shown that antimicrobial agents and NPs can induce modifications in outer membrane components, influencing bacterial immune evasion strategies [65,66].

In terms of adherence, *P. aeruginosa* regulated such proteins as ChpB, CheW, FimL, FimX, RpoN, RpoS, PilB, PilH, PilI, and PilU, all of which contribute to bacterial attachment and colonization [67]. Similarly, *P. mirabilis* exhibited differential expression of adherence-associated proteins, including Crp, MatB, MetQ, MrfD, Tuf, and Uca. The presence of multiple adherence-related proteins in both bacteria suggests that Se NP exposure may influence bacterial attachment properties, possibly altering surface interactions that impact colonization and biofilm formation. Biofilm-associated proteins were also significantly affected in both species. *P. aeruginosa* altered the expression of AlgB, AlgP, AlgR, LasR, and RhlR, which are key regulators of alginate biosynthesis and quorum sensing, crucial for biofilm maintenance [67]. In contrast, *P. mirabilis* differentially expressed such biofilm-related proteins as KGA92257.1 and RpoE. These findings suggest that Se NP exposure may disrupt bacterial aggregation and biofilm maturation, potentially weakening bacterial persistence mechanisms. Previous studies have demonstrated that silver NPs can interfere with bacterial adhesion and biofilm formation by disrupting cell surface structures and altering quorum-sensing pathways [68].

Secretion system proteins were altered in both species, indicating an increased emphasis on protein translocation and virulence factor secretion under stress. In *P. aeruginosa*, several type II and type III secretion system components, including GspE, XcpR-S, and YebC [67], were differentially expressed. Similarly, *P. mirabilis* exhibited differential expression of secretion system-related proteins, including Hcp, ImpB, Impl, PpiA, and TssQ_1, components associated with the type VI secretion system [64]. These findings suggest that both bacteria respond to Se NP stress by modulating secretion system components, possibly to facilitate the export of virulence factors or stress response proteins. Similar trends have been observed in *Vibrio parahaemolyticus* and *S. aureus*, where exposure to antimicrobial agents led to the modulation of secretion systems, aiding in bacterial persistence [69,70].

Immune modulation-related proteins exhibited distinct patterns between the two species. *P. aeruginosa* showed regulation of Gmd, LpxK, RfaG, RfbA, and RhlA, proteins involved in immune evasion and bacterial defense [67]. In contrast, *P. mirabilis* displayed a broader immune modulation response, with differentially expressed proteins including AcpP, GalE, GlmU, KdsA, KdtA, LpxB-C, LpxL, MsbA, RfaC, and WecC. Many of these motility-associated proteins were widely affected in both bacteria, indicating a potential impact on bacterial movement and environmental navigation under stress. In *P. aeruginosa*, proteins such as CheW, DctD, FlaG, FleR, FlgD-G, FlgL, FlgN, FliC, FliF, FliH, MotA, MotD, and NtrC were differentially expressed, while *P. mirabilis* exhibited regulation of motility-related proteins, including CheZ, FlgB, FlgF-H, FlgJ, FlgM, FliG-H, FliM, GlnG, Tap, and YfhA. These findings indicate that both species may be modulating their flagellar and chemotactic systems in response to Se NP exposure.

A key distinction between the two species was the differential regulation of stress survival proteins. *P. mirabilis*, but not *P. aeruginosa*, exhibited differential regulation of ClpB, KatA, SodC, and UreC, all of which are associated with bacterial stress survival and oxidative stress resistance [64]. This suggests that *P. mirabilis* may rely on oxidative stress resistance mechanisms to counteract Se NP toxicity, likely through increased antioxidant enzyme activity and proteolysis of damaged proteins. In contrast, the absence of similar stress survival proteins in *P. aeruginosa* may indicate that this species mitigates oxidative stress through alternative pathways, such as enhanced energy metabolism or efflux system activation.

Selenium nanoparticles are well known to catalyze the generation of reactive oxygen species, which impose oxidative stress and damage key biomolecules including DNA, proteins, and lipids. Han et al. demonstrated that Se NPs induce intracellular reactive oxygen species accumulation in methicillin-resistant *S. aureus* and vancomycin-resistant *Enterococcus*, ultimately triggering apoptosis-like bacterial death [13]. Consistent with this, our proteomic analysis revealed upregulation of stress response proteins, further supporting reactive oxygen species-mediated oxidative stress as a central mechanism of Se NP antibacterial activity.

### 3.7. Differentially Expressed Proteins Associated with Antimicrobial Resistance

The analysis of differentially expressed proteins associated with antimicrobial resistance in *P. aeruginosa* and *P. mirabilis* following Se NP treatment revealed distinct regulatory patterns, indicating species-specific strategies for mitigating NP-induced stress and potential antimicrobial effects (Table 4 and Table 5). In *P. aeruginosa*, 10 antimicrobial resistance-related proteins were differentially expressed, including efflux system components (AdeR, MexG, MexH, MexL, MexV, OpmD, RsmA) and antibiotic target replacement proteins (DfrA3, MyrA, RpoB). In *P. mirabilis*, 10 antimicrobial resistance-associated proteins were also differentially expressed, including efflux-related transporters (CpxA, CpxR, Crp, Hns, LptD, MsbA, RsmA, TolC), an antibiotic target alteration regulator (PhoB), and an antibiotic-modifying enzyme (CatA4). The expression patterns suggest that both bacteria employ efflux-mediated resistance and structural modifications to counteract Se NP-induced stress.

Efflux pump-associated proteins were regulated in both species, suggesting that active transport mechanisms play a central role in bacterial adaptation to Se NP exposure. In *P. aeruginosa*, MexG, MexH, MexV, and OpmD—components of the MexGHI-OpmD efflux system—were differentially expressed, indicating increased activity of the resistance-nodulation-cell division (RND) efflux system [71]. These efflux pumps are known to contribute to multidrug resistance by expelling toxic compounds, including antimicrobials and heavy metals, from Gram-negative bacteria. Additionally, the MexGHI-OpmD efflux system has been reported to be involved in stress responses in *P. aeruginosa*, further supporting its role in counteracting NP-induced toxicity [71].

Similarly, *P. mirabilis* displayed overexpression of TolC, MsbA, and LptD, components of tripartite efflux pumps and lipopolysaccharide-assembly pathways [72]. These proteins facilitate resistance against antimicrobial agents, metal ions, and environmental stressors. The upregulation of these efflux-related proteins suggests that *P. mirabilis* employs efflux-mediated detoxification to reduce intracellular accumulation of Se NPs, a common strategy observed in bacteria exposed to heavy metals and antibiotics [73,74].

Regulatory proteins governing antimicrobial resistance and stress responses were also differentially expressed in both species, indicating complex transcriptional and post-transcriptional regulation in response to Se NPs. In *P. aeruginosa*, RsmA, AdeR, and MexL were differentially expressed. RsmA plays a crucial role in secondary metabolism and antimicrobial resistance by modulating RNA stability and translation [75]. AdeR, a well-known regulator of efflux pump expression, facilitates adaptive resistance by modulating bacterial responses to environmental stress [76]. The observed downregulation of MexL, a negative regulator of the MexXY efflux system, suggests that Se NP exposure may trigger the derepression of certain efflux pumps, possibly leading to increased antimicrobial resistance [72].

### 3.8. Differentially Expressed Proteins Associated with Heavy Metal Resistance

The analysis of differentially expressed proteins associated with heavy metal resistance in *P. aeruginosa* and *P. mirabilis* following Se NP treatment revealed distinct regulatory patterns, reflecting species-specific strategies for coping with metal-induced stress (Table 6 and Table 7). *P. aeruginosa* exhibited differential expression of 17 heavy metal resistance-related proteins, including metal transporters (CorA, CorD, CueA), mercury detoxification proteins (MerA, MerP, MerR), regulatory transcription factors (CopR, MexT, ComR, CpxR, RpoS), and stress tolerance proteins (NirD/YgiW/YdeI family). In contrast, *P. mirabilis* displayed a broader response, with 25 differentially expressed heavy metal resistance proteins, including tellurite resistance proteins (TerB, TerD, TehB), phosphate transporters (PitA, PstB, PstS), copper homeostasis proteins (CutC, CuiD), metal-binding proteins (ZinT, ZntA), and regulatory proteins (ModE, PcoR, CpxR, IclR). These findings suggest that while both species employ metal transport, detoxification, and regulatory mechanisms, *P. mirabilis* activates a more extensive set of resistance pathways to counteract Se NP-induced toxicity.

Differential expression of metal transporters in both *P. aeruginosa* and *P. mirabilis* highlights the crucial role of metal ion homeostasis in their adaptive responses to Se NP stress. In *P. aeruginosa*, the expression of CorA (ion transporter) and CorD (a Co^2+^/Mg^2+^ efflux protein) was modulated, indicating a tightly controlled mechanism for metal ion homeostasis. Additionally, CueA, a copper-translocating P-type ATPase, was differentially expressed, suggesting that *P. aeruginosa* employs active copper transport mechanisms to mitigate metal toxicity. These findings are consistent with previous reports showing that CueA contributes to metal homeostasis and resistance in *P. aeruginosa* exposed to metal stress [77].

In *P. mirabilis*, multiple transport proteins were modulated, including CutC (copper homeostasis protein), CuiD (blue copper oxidase), ZinT (metal-binding protein), and ZntA (cadmium-translocating P-type ATPase). CutC and CuiD are known to contribute to bacterial copper tolerance, potentially by modulating copper oxidation and transport [78]. Meanwhile, ZinT and ZntA contribute to resistance against multiple heavy metals, including zinc, lead, cobalt, mercury, and cadmium, suggesting that *P. mirabilis* employs a wider range of transporters to maintain intracellular metal ion balance [79].

A notable difference between the two species was the activation of mercury resistance proteins in *P. aeruginosa*. Mer operon components MerA (Mercury(II) reductase), MerP (periplasmic binding protein), and MerR (mercury resistance transcriptional regulator) were differentially expressed. The MerA enzyme plays a crucial role in reducing toxic mercury (Hg^2+^) to volatile elemental mercury (Hg^0^), which is then expelled from the cell [80]. Presence of these detoxification proteins suggests that Se NP exposure may regulate heavy metal detoxification pathways in *P. aeruginosa*.

Conversely, *P. mirabilis* did not exhibit mercury resistance protein expression but instead displayed upregulation of tellurite resistance proteins (TerB, TerD, TehB). Tellurite resistance has been linked to oxidative stress protection, suggesting that *P. mirabilis* may rely on tellurite resistance mechanisms rather than mercury detoxification to mitigate NP-induced stress [81]. The upregulation of tellurite resistance proteins supports previous findings that tellurite resistance pathways enhance bacterial survival under oxidative and metal stress conditions [82].

Differential expression of several membrane-associated proteins suggests that both species undergo structural remodeling of their outer membranes to adapt to Se NP-induced stress. In *P. aeruginosa*, ModC (a heme ABC transporter ATP-binding protein) and VcaM (a lipid A export permease/ATP-binding protein) were differentially expressed, potentially indicating modifications in membrane permeability and lipid transport. Lipid A modifications have been shown to contribute to bacterial resistance against metal toxicity by reducing membrane permeability to toxic ions [83].

In *P. mirabilis*, LptD (lipopolysaccharide-assembly protein) and PitA (a low-affinity phosphate transporter) were differentially expressed, further supporting the hypothesis that membrane integrity and phosphate metabolism play crucial roles in stress adaptation. Increased expression of LptD suggests that *P. mirabilis* may reinforce its lipopolysaccharide layer to mitigate Se NP-induced damage, a strategy previously observed in bacteria exposed to metal stress [84].

A limitation of this study is that we conducted proteomic analysis only on *P. aeruginosa* and *P. mirabilis*, which exhibited the highest and lowest susceptibility to Se NPs, respectively. While this approach provided valuable insights into species-specific adaptive responses, it did not capture the full spectrum of proteomic changes across all tested uropathogens, including *E. coli* and *E. faecalis*. As a result, potential differences in stress adaptation mechanisms among other bacterial species remain unexplored. Future studies should expand proteomic analysis over a broader range of pathogens to provide a more comprehensive understanding of bacterial responses to Se NP-induced stress.

Another consideration is the potential loss of proteins during sample processing. Since Se NPs induce cell membrane damage, proteins—particularly secreted proteins or intracellular proteins released due to cell rupture—may leak into the surrounding medium and be lost during the centrifugation step post-treatment. This means that the detected proteomic profile primarily reflects proteins retained within partially intact or damaged cells and may not represent the full proteome. Furthermore, differences in proteomic profiles between *P. aeruginosa* and *P. mirabilis* could partly be attributed to differences in cellular integrity, as *P. aeruginosa* is more susceptible and prone to lysis, whereas *P. mirabilis* is more resistant. Despite this limitation, the detection of differentially regulated proteins provides meaningful insight into bacterial stress responses and adaptation strategies following Se NP exposure.

Finally, due to budgetary constraints, proteome analyses were not conducted in triplicates. Performing proteomic profiling for two bacterial species across three time points would have required substantial additional resources beyond the scope of the current funding. While this represents a limitation in statistical replication, we minimized variability by applying stringent data processing, normalization, and filtering criteria. The observed pathway-level trends were consistent with established bacterial stress responses, lending confidence to the robustness of the findings. Future studies, contingent on available funding, should aim to include triplicate analyses to further enhance the statistical power and reproducibility of the proteomic data.

## 4. Conclusions

This study provides comprehensive insights into the antibacterial mechanisms of Se NPs against uropathogenic bacteria. Antibacterial assays demonstrated that Se NPs effectively inhibited bacterial growth, though susceptibility varied among species. Our findings revealed that Se NPs induce significant morphological alterations and trigger distinct proteomic responses in uropathogens. While both *P. aeruginosa* and *P. mirabilis* modulated core cellular functions, including energy metabolism and transcriptional regulation, *P. mirabilis* displayed a more pronounced upregulation of proteins involved in cell wall biogenesis. The upregulation of two-component systems, chemotaxis, flagellar assembly, and ABC transporters in both species underscores the importance of environmental sensing, motility, and transmembrane transport in mitigating NP toxicity. Furthermore, analysis of virulence, antimicrobial resistance, and heavy metal resistance proteins revealed species-specific adaptations, including differential regulation of detoxification pathways. These results accentuate the complexity of bacterial responses to Se NP exposure and emphasize the importance of considering species-specific adaptations when evaluating their potential as antimicrobial agents.

## Figures and Tables

**Figure 1 nanomaterials-15-01404-f001:**
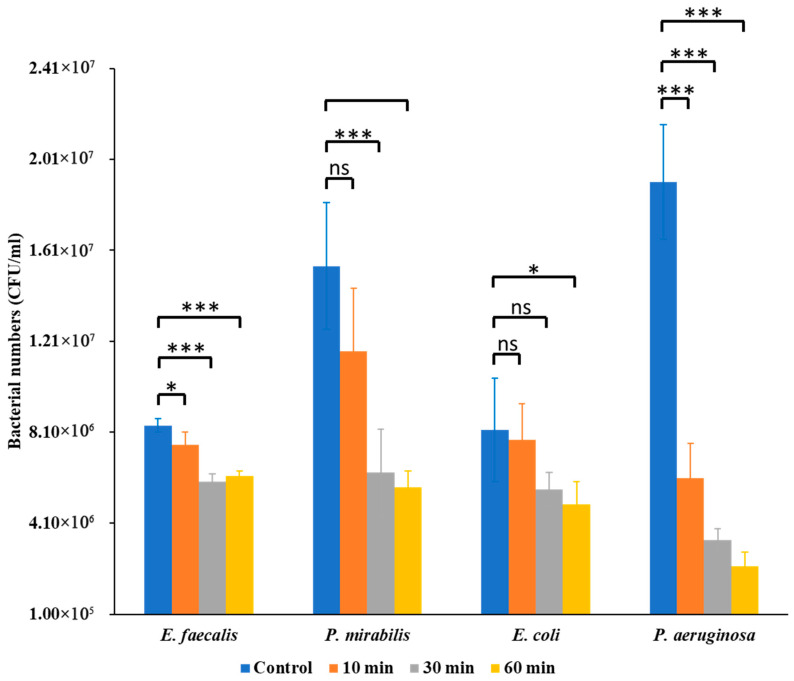
Bactericidal activities of Se NPs on *E. faecalis* ATCC 29212, *P. mirabilis* ATCC 7002, *E. coli* ATCC 700928, and *P. aeruginosa* PA14. A single asterisk (*) indicates statistically significant differences between groups at *p* < 0.05, while three (***) asterisks denote increasingly stringent significance level (*p* < 0.001).

**Figure 2 nanomaterials-15-01404-f002:**
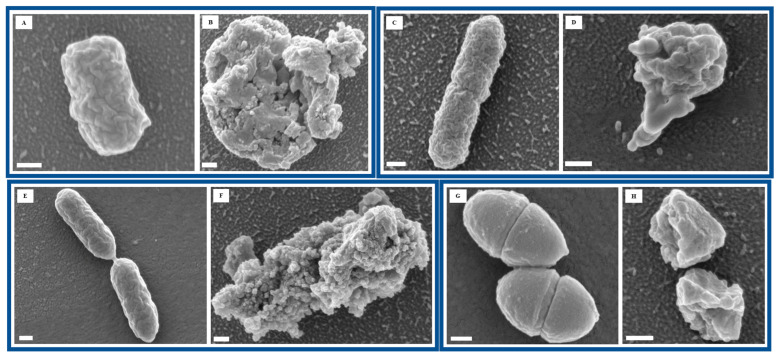
FESEM images of *E. coli* ATCC 700928 (**A**,**B**), *P. mirabilis* ATCC 7002 (**C**,**D**), *P. aeruginosa* PA14 (**E**,**F**), and *E. faecalis* ATCC 29212 (**G**,**H**). The scale bar in the images corresponds to 200 nm. (**A**,**C**,**E**,**G**) are untreated (control), and (**B**,**D**,**F**,**H**) are Se NP-treated for 60 min.

**Figure 3 nanomaterials-15-01404-f003:**
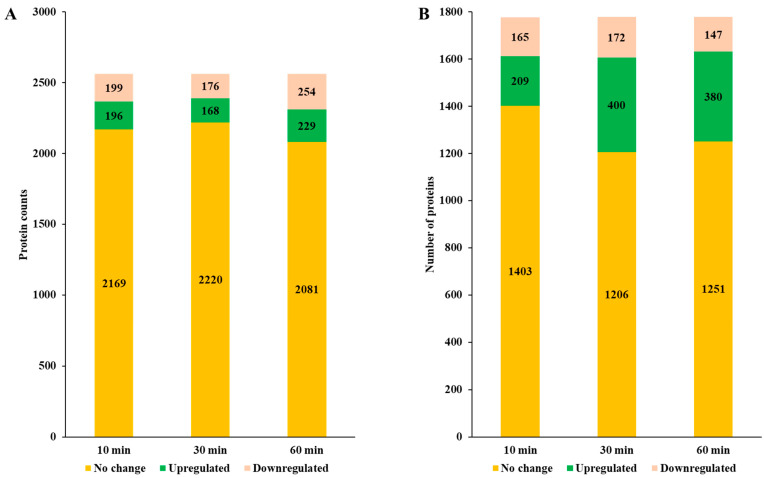
Number of differentially expressed proteins in *P. aeruginosa* PA14 (**A**) and *P. mirabilis* ATCC 7002 (**B**).

**Figure 4 nanomaterials-15-01404-f004:**
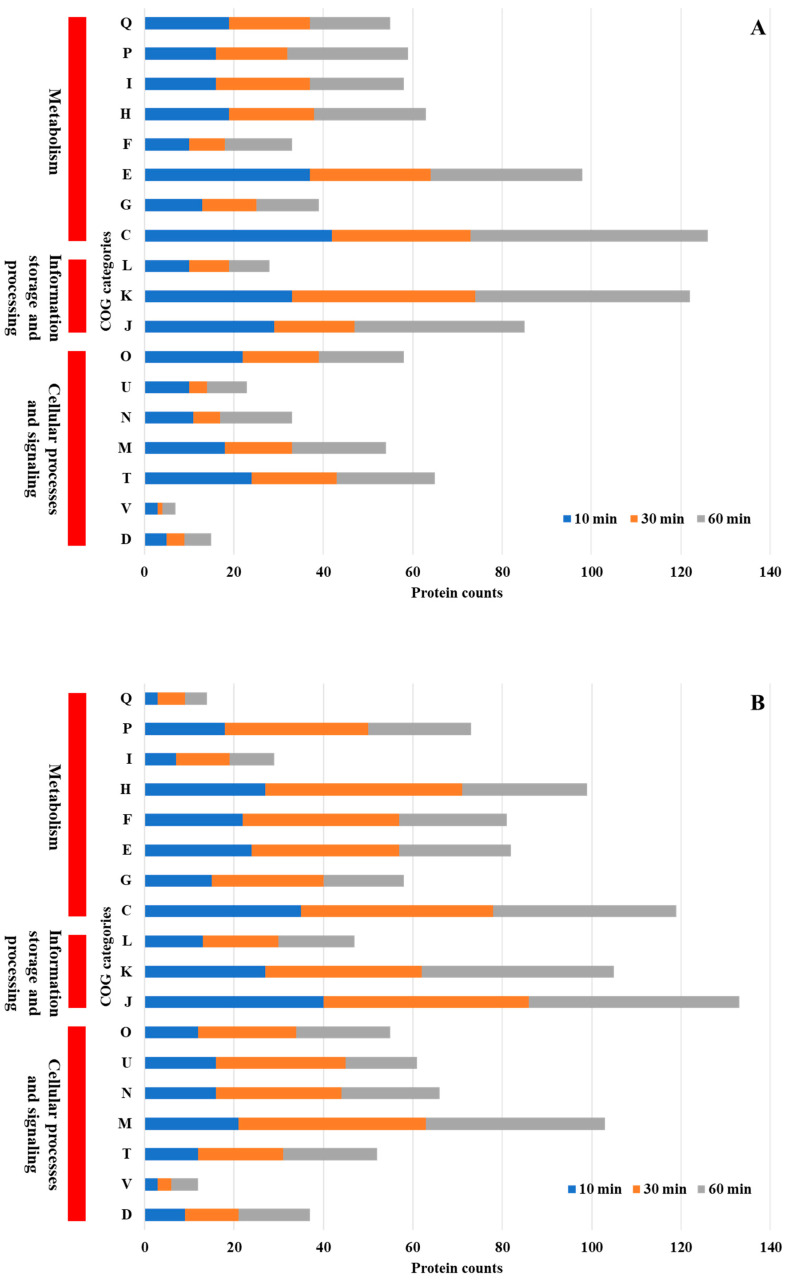
COG functional classification of differentially expressed proteins in *P. aeruginosa* PA14 (**A**) and *P. mirabilis* ATCC 7002 (**B**). COG functional categories: J, translation, ribosomal structure, and biogenesis; K, transcription; L, replication, recombination, and repair; D, cell cycle control, cell division, chromosome partitioning; V, defense mechanisms; T, signal transduction mechanisms; M, cell wall/membrane/envelope biogenesis; N, cell motility; U, intracellular trafficking, secretion, and vesicular transport; O, post-translational modification, protein turnover, chaperones; C, energy production and conversion; G, carbohydrate transport and metabolism; E, amino acid transport and metabolism; F, nucleotide transport and metabolism; H, coenzyme transport and metabolism; I, lipid transport and metabolism; P, inorganic ion transport and metabolism; Q, secondary metabolite biosynthesis, transport, and catabolism. Poorly characterized (S) was excluded.

**Figure 5 nanomaterials-15-01404-f005:**
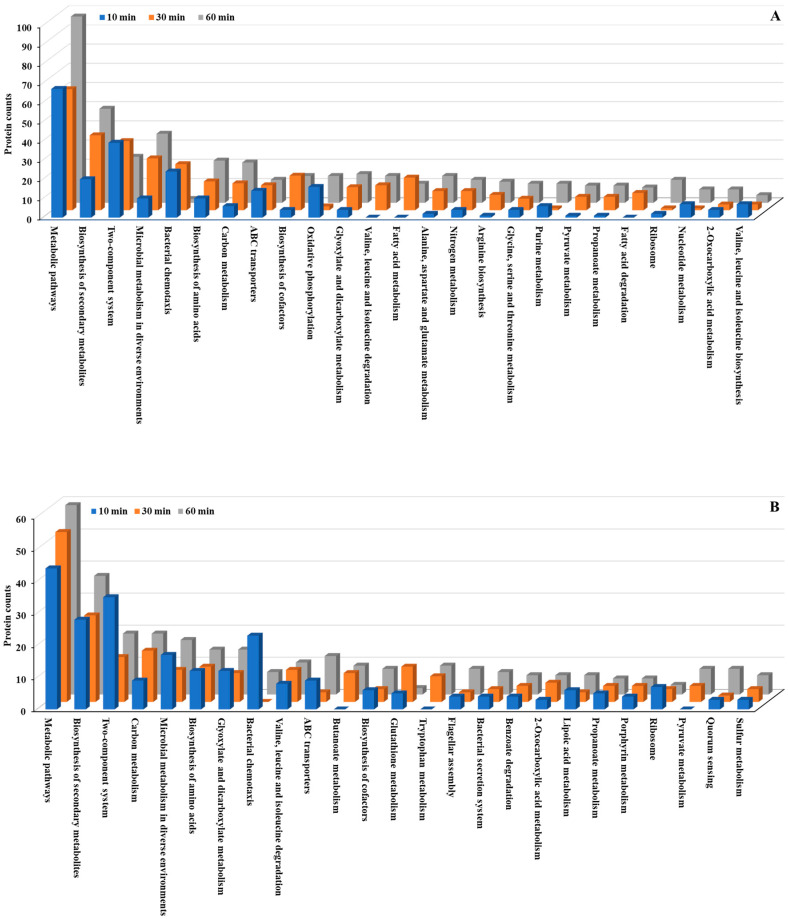
Top 25 KEGG pathways in differentially expressed proteins identified from *P. aeruginosa* PA14. (**A**): upregulated proteins, (**B**): downregulated proteins.

**Figure 6 nanomaterials-15-01404-f006:**
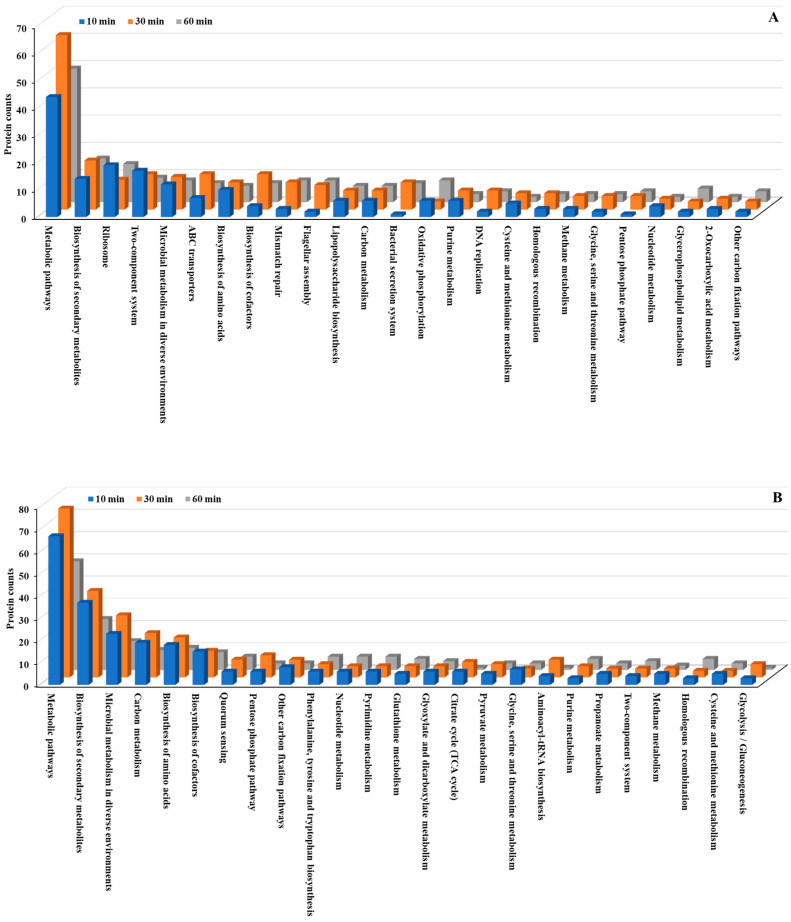
Top 25 KEGG pathways in differentially expressed proteins identified from *P. mirabilis* ATCC 7002. (**A**): upregulated proteins, (**B**): downregulated proteins.

**Figure 7 nanomaterials-15-01404-f007:**
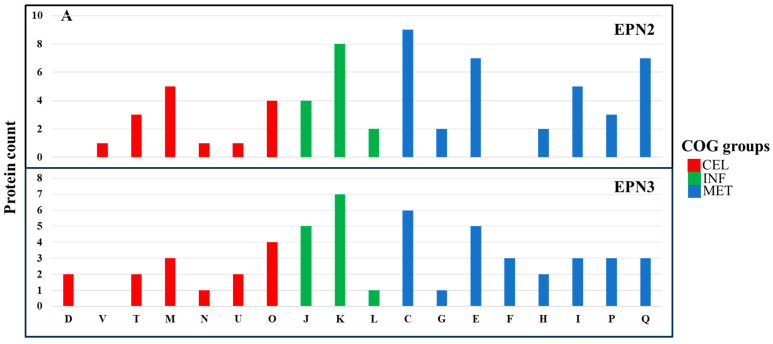
Protein expression patterns (EPNs) and functional distribution of proteins showing *P. aeruginosa* PA14 (**A**) and *P. mirabilis* ATCC 7002 (**B**). CEL (cellular process and signaling), INF (information storage and processing), and MET (metabolism). COG functional categories: J, translation, ribosomal structure, and biogenesis; K, transcription; L, replication, recombination, and repair; D, cell cycle control, cell division, chromosome partitioning; V, defense mechanisms; T, signal transduction mechanisms; M, cell wall/membrane/envelope biogenesis; N, cell motility; U, intracellular trafficking, secretion, and vesicular transport; O, post-translational modification, protein turnover, chaperones; C, energy production and conversion; G, carbohydrate transport and metabolism; E, amino acid transport and metabolism; F, nucleotide transport and metabolism; H, coenzyme transport and metabolism; I, lipid transport and metabolism; P, inorganic ion transport and metabolism; Q, secondary metabolite biosynthesis, transport, and catabolism.

**Table 1 nanomaterials-15-01404-t001:** Nanotechnology-enabled therapeutic strategies for the treatment and prevention of UTIs.

Nanoparticles	Sizes	Urinary Bacteria	Key Activity	Reference
Zinc Oxide	70.9 ± 3.0 nm	*S. epidermidis*, *S. aureus*, *E. coli*, *K. pneumoniae*, *P. mirabilis*, *P. aeruginosa*	Antibiofilm	[16]
Zinc Oxide	60 nm	*E. faecalis*	Antibacterial and antibiofilm	[17]
Silver	50–60 nm	*E. coli*, *P. aeruginosa*	Antibacterial and antibiofilm	[18]
Silver	26–72 nm	*E. faecalis*, *E. coli*, *S. aureus*	Antibacterial	[19]
Silver	126.3 nm	*P. mirabilis*	Antibacterial and antibiofilm	[20]
Silver	28 to 60 nm	*E. coli*, *K. pneumoniae*, *A. baumannii*, *P. aeruginosa*, *P. mirabilis*, *E. faecalis*, *S. arlettae*	Antibacterial	[21]
Gold	50–60 nm	*E. coli*, *K. pneumoniae*, *P. vulgaris*, *A. baumannii*, *S. aureus*, *E. faecalis*	Antibacterial	[22]
Silver	20–120 nm	*E. faecalis*, *S. aureus*, *S. epidermidis*, *S. saprophyticus*, *E. coli*, *K. pneumoniae*, *P. mirabilis*, *P. aeruginosa*, *C. albicans*	Antibiofilm	[23]
Silver	10–60 nm	*E. coli*, *K. pneumoniae*, *P. aeruginosa*, *S. saprophyticus*	Antibacterial	[24]
Silver	55.7 nm	*E. faecalis*	Antibiofilm	[25]
Silver	14 nm	*S. aureus*, *E. coli*, *P. aeruginosa*	Antibacterial and antibiofilm	[26]
Copper	5–20 nm	*E. coli*, *Enterococcus* sp., *Proteus* sp., *Klebsiella* sp.	Antibacterial	[27]
Silver	20–400 nm	*S. marcescens*, *P. mirabilis*	Antibiofilm	[28]
Zinc Oxide	20–40 nm	*C. albicans*	Antibiofilm	[29]
Copper	48 ± 4 nm	*E. coli*	Antibacterial	[30]

**Table 2 nanomaterials-15-01404-t002:** Differentially expressed proteins associated with virulence in *P. aeruginosa* PA14.

Protein ID	Protein Name	Gene Name	COG	Fold Ratio		
				10 min	30 min	60 min
WIV37158.1	Flagellar hook assembly protein	flgD	N	2.17	2.50	2.47
WIV38668.1	Alginate biosynthesis regulator	algR	KT	1.06	1.02	1.15
WIV37269.1	Transcriptional regulatory protein	yebC	K	−1.11	−1.21	−1.73
WIV40961.1	Tetraacyldisaccharide 4′-kinase	lpxK	F	−1.08	−1.19	−1.31
WIV37818.1	Type IV-A pilus assembly ATPase	pilB	NU	1.03	0.93	1.11
WIV38362.1	Cyclic di-GMP-binding protein	fimX	T	1.04	0.94	1.37
WIV38414.1	Glycosyltransferase family 4 protein	rfaG	M	1.01	0.85	1.19
WIV36768.1	Flagellar motor protein	motB	N	−1.09	−0.57	−1.03
WIV40575.1	Flagellar export chaperone	flgN	NOU	−1.06	−0.52	−1.56
WIV36404.1	Type IV pilus/biofilm regulator	fimL	T	0.44	0.06	1.50
WIV36767.1	ParA family protein	-	D	0.03	0.33	1.02
WIV37136.1	Flagellar basal-body MS-ring/collar protein	fliF	N	0.77	0.52	1.24
WIV37138.1	Sigma-54-dependent response regulator transcription factor	fleR	T	−0.90	0.33	1.70
WIV37156.1	Flagellar basal-body rod protein	flgF	N	0.89	0.63	1.17
WIV38357.1	Flagellar motor stator protein	motA	N	0.78	0.58	1.15
WIV38857.1	GDP-mannose 4,6-dehydratase	gmd	M	0.69	0.76	1.02
WIV39375.1	Type IV pilus ATPase	pilU	NU	−0.29	−0.56	1.11
WIV39389.1	Chemotaxis protein	pilI	NT	0.76	0.70	1.14
WIV40448.1	Transcriptional regulator	sdiA	K	0.64	0.70	1.26
WIV38569.1	Glucose-1-phosphate thymidylyltransferase	rfbA	M	1.13	0.48	0.63
WIV38885.1	Sigma-54-dependent response regulator transcription factor	algB	T	1.06	0.24	−0.73
WIV39703.1	Pyocyanin biosynthetic protein	mhbM	CH	1.32	−0.03	−0.37
WIV40032.1	Nitrate reductase subunit beta	narH	C	1.14	0.94	0.60
WIV40031.1	Nitrate reductase subunit alpha	narG	C	1.08	0.74	0.98
WIV36765.1	Chemotaxis protein	cheW	NT	1.03	1.33	0.99
WIV37144.1	Flagellar protein	flaG	N	−0.17	−0.69	−1.07
WIV37150.1	Flagellar hook-associated protein	flgL	N	0.20	−0.46	−1.32
WIV37155.1	Flagellar basal-body rod protein	flgG	N	−0.39	−0.38	−1.02
WIV37303.1	TonB-dependent siderophore receptor	fepA	P	−0.85	−0.73	−1.14
WIV37329.1	Carbon storage regulator	csrA	J	−0.34	−0.90	−1.52
WIV38529.1	Two-component system response regulator	glnG	T	−0.82	−0.85	−1.23
WIV38660.1	Alginate regulator	algP	C	−0.66	−0.52	−1.33
WIV39699.1	Fe(3+)-pyochelin receptor	fptA	P	−0.33	−0.75	−1.02
WIV40843.1	GspF family T2SS innner membrane protein variant	gspF	U	−0.29	−0.19	−1.76
WIV36799.1	Transcriptional regulator	lasR	K	−2.07	−0.81	−0.56
WIV37134.1	Flagellar assembly protein	fliH	N	−1.11	−0.54	−0.76
WIV39393.1	Chemotaxis protein	chpB	NT	−1.02	−0.89	−0.72
WIV40321.1	RNA polymerase sigma factor	rpoS	K	−1.15	−0.19	0.12
WIV40446.1	3-(3-hydroxydecanoyloxy)decanoate synthase	rhlA	I	−1.00	−0.63	−0.50
WIV40842.1	Type II secretion system protein E	gspE	NU	−1.74	0.60	0.43
WIV36716.1	Ureidoglycolate lyase	allA	F	−0.47	−1.17	−1.07
WIV37145.1	B-type flagellin	fliC	N	−0.48	−2.28	−3.84
WIV37157.1	Flagellar hook protein	flgE	N	−0.47	−1.63	−2.51
WIV37754.1	RNA polymerase factor sigma-54	rpoN	K	−0.96	−1.75	−1.62
WIV38911.1	Sigma-54 dependent transcriptional regulator	dctD	T	−0.79	−1.02	−1.04
WIV39388.1	Twitching motility response regulator	pilH	KT	−0.79	−1.02	−1.19

**Table 3 nanomaterials-15-01404-t003:** Differentially expressed proteins associated with virulence in *P. mirabilis* ATCC 7002.

Protein ID	Protein Name	Gene Name	COG	Fold Ratio		
				10 min	30 min	60 min
KGA91932.1	Hypothetical protein DR94_2402	-	NT	1.14	1.54	1.60
KGA89703.1	3-deoxy-8-phosphooctulonate synthase	kdsA	F	−1.69	−1.86	−1.15
KGA90333.1	Hypothetical protein DR94_3439	-	S	1.41	0.95	1.29
KGA91088.1	Flagellar basal-body rod protein	flgF	N	1.30	0.70	2.64
KGA89142.1	Isocitrate lyase	aceA	C	−1.54	−1.24	−0.82
KGA89261.1	Translation elongation factor Tu	tuf	J	−1.58	−1.67	−0.45
KGA90858.1	UTP-glucose-1-phosphate uridylyltransferase	-	JM	−1.48	−1.38	−0.93
KGA91915.1	Bacterial regulatory s, crp family protein	crp	K	−1.08	−1.55	−0.78
KGA92002.1	Translation elongation factor Tu	tuf	J	−1.46	−2.50	−0.52
KGA92112.1	Nucleotide sugar dehydrogenase family protein	wecC	C	−1.15	−1.43	−0.91
KGA90032.1	FHA domain protein	impI	T	0.20	1.36	0.88
KGA90576.1	AAA domain family protein	yfhA	T	0.77	1.06	0.77
KGA90599.1	RNA polymerase sigma factor	rpoE	K	−0.15	1.73	0.19
KGA90670.1	Flagellar L-ring family protein	flgH	N	0.66	1.10	0.61
KGA90924.1	Catalase	katA	P	0.57	1.11	−0.22
KGA91197.1	Flagellar M-ring protein	fliF	NU	0.55	1.17	0.63
KGA91239.1	Flagellar motor switch protein	fliG	N	0.48	1.06	0.67
KGA92049.1	Peptidyl-prolyl cis-trans isomerase A	ppiA	M	0.22	1.02	0.72
KGA92300.1	Aspartate carbamoyltransferase	pyrB	F	0.27	1.08	0.14
KGA92373.1	Glycosyl transferases group 1 family protein	kdtA	H	−0.28	1.18	0.12
KGA88983.1	Lipid-A-disaccharide synthase	lpxB	M	−2.11	−1.71	1.33
KGA92257.1	Response regulator	-	KT	1.74	0.35	−1.42
KGA91588.1	Carbamoyl-phosphate synthase large chain	carB	F	−0.50	−1.11	−1.13
KGA91890.1	Nitrogen regulation protein NR	glnG	T	0.83	−1.42	−3.68
KGA92042.1	Type VI secretion system effector, Hcp1 family protein	hcp	S	−0.15	−1.23	−1.04
KGA89047.1	D-methionine-binding lipoprotein	metQ	M	0.49	1.18	1.60
KGA90253.1	Hypothetical protein DR94_2357	impB	S	0.68	1.30	1.36
KGA90581.1	Lipid A biosynthesis lauroyl acyltransferase	lpxL	M	−0.40	1.03	1.05
KGA90704.1	Flagellar basal-body rod protein	flgG	N	0.57	1.32	2.11
KGA90822.1	Flagellar basal-body rod protein	flgB	N	0.55	1.46	1.96
KGA90964.1	Flagellar rod assembly protein/muramidase	flgJ	MNOU	-	9.97	9.97
KGA91119.1	Flagellar P-ring family protein	-	T	0.54	1.32	2.11
KGA91216.1	Flagellar assembly FliH family protein	fliH	N	0.99	2.60	1.72
KGA91265.1	Hypothetical protein DR94_351	tap	NT	0.68	1.25	1.60
KGA91966.1	Fimbrillin	matB	S	0.55	2.26	1.94
KGA92160.1	dTDP-glucose 4,6-dehydratase	-	G	−1.20	−0.39	−0.78
KGA90476.1	Nitrate reductase, alpha subunit	narG	C	1.63	0.70	−0.30
KGA89485.1	Acyl carrier protein	acpP	IQ	0.63	0.81	1.61
KGA89838.1	Superoxide dismutase Cu-Zn	sodC	P	0.55	0.89	2.48
KGA90056.1	Fimbrial family protein	uca	NU	0.51	1.00	2.29
KGA90208.1	Lipid A export permease/ATP-binding protein	msbA	V	0.05	0.87	1.44
KGA90291.1	Carbon storage regulator	csrA	J	−0.11	0.92	1.41
KGA90582.1	Flagellar motor switch protein	fliM	N	−0.42	0.99	2.09
KGA90677.1	Flagellar hook protein	flgE	S	0.53	0.92	2.13
KGA91171.1	Protein phosphatase	cheZ	NT	−0.19	0.36	1.39
KGA91580.1	Type VI secretion system effector, Hcp1 family protein	tssQ_1	S	−0.83	−0.46	1.91
KGA91834.1	Hypothetical protein DR94_2403	-	NT	0.10	0.84	1.44
KGA92484.1	Lipopolysaccharide heptosyltransferase I	rfaC	M	0.08	0.52	1.34
KGA90317.1	ATP-dependent chaperone protein	clpB	O	−0.87	−1.04	−0.42
KGA90503.1	UDP-glucose 4-epimerase	galE	M	−0.73	−1.18	0.00
KGA88936.1	Beta-hydroxyacyl-(acyl-carrier-protein) dehydratase	fabZ	I	1.24	1.11	0.44
KGA90544.1	UDP-3-O-[3-hydroxymyristoyl] N-acetylglucosamine deacetylase	lpxC	F	1.46	3.74	−0.26
KGA91159.1	TonB-dependent hemoglobin/transferrin/lactoferrin receptor family protein	hmuR	P	1.14	1.37	0.40
KGA91541.1	Hypothetical protein DR94_1982	mrfD	NU	1.68	2.97	−0.01
KGA91697.1	Urease, alpha subunit	ureC	E	1.12	1.06	0.20
KGA91802.1	Hypothetical protein DR94_1992	mrfD	NU	1.19	2.08	0.85
KGA90440.1	Nitrate reductase, beta subunit	narH	C	0.84	0.78	−1.30
KGA90968.1	Capsular synthesis regulator component B	rcsB	K	0.21	0.02	−1.39
KGA92370.1	UDP-N-acetylglucosamine diphosphorylase/glucosamine-1-phosphate N-acetyltransferase	glmU	M	−0.59	−0.87	−1.33

**Table 4 nanomaterials-15-01404-t004:** Differentially expressed proteins associated with antimicrobial resistance in *P. aeruginosa* PA14.

Protein ID	Protein Name	Gene Name	COG	Fold Ratio		
				10 min	30 min	60 min
WIV37329.1	Carbon storage regulator	rsmA	J	−0.34	−0.90	−1.52
WIV38764.1	Phosphate regulon transcriptional regulator	adeR	K	−0.96	−0.92	−1.56
WIV39713.1	Multidrug efflux RND transporter inhibitory subunit	mexG	S	1.18	−0.22	−0.03
WIV39712.1	Multidrug efflux RND transporter periplasmic adaptor	mexH	M	0.60	0.78	1.01
WIV37667.1	Multidrug efflux RND transporter periplasmic adaptor subunit	mexV	M	0.19	2.20	−0.20
WIV39329.1	Dihydrofolate reductase	dfrA3	H	−0.51	1.02	0.46
WIV37075.1	Methyltransferase domain-containing protein	myrA	Q	−1.57	−1.05	−0.57
WIV39710.1	Multidrug efflux transporter outer membrane subunit	opmD	MU	1.36	−0.43	−1.20
WIV39651.1	DNA-directed RNA polymerase subunit beta	rpoB	K	1.09	0.63	1.05
WIV40266.1	Efflux system transcriptional repressor	mexL	K	−1.42	−1.08	−1.38

**Table 5 nanomaterials-15-01404-t005:** Differentially expressed proteins associated with antimicrobial resistance in *P. mirabilis* ATCC 7002.

Protein ID	Protein Name	Gene Name	COG	Fold Ratio		
				10 min	30 min	60 min
KGA90065.1	Chloramphenicol acetyltransferase	catA4	H	1.66	1.62	−0.19
KGA90208.1	Lipid A export permease/ATP-binding protein	msbA	V	0.05	0.87	1.44
KGA90291.1	Carbon storage regulator	rsmA	J	−0.11	0.92	1.41
KGA90545.1	DNA-binding protein H-NS	hns	K	0.80	0.87	1.79
KGA91504.1	Phosphate regulon transcriptional regulatory protein	phoB	K	0.51	0.02	2.61
KGA92342.1	Sensor protein	cpxA	T	−0.75	−0.39	3.75
KGA89035.1	Outer membrane protein	tolC	S	0.62	1.24	0.13
KGA92341.1	Transcriptional regulatory protein	cpxR	K	0.38	1.03	0.07
KGA91915.1	Catabolite activator protein	crp	K	−1.08	−1.55	−0.78
KGA89026.1	LPS-assembly protein	lptD	M	1.40	1.55	1.18

**Table 6 nanomaterials-15-01404-t006:** Differentially expressed proteins associated with heavy metal resistance in *P. aeruginosa* PA14.

Protein ID	Protein Name	Gene Name	COG	Fold Ratio		
				10 min	30 min	60 min
WIV38676.1	Ion transporter	corA	P	0.53	−1.19	−1.51
WIV40321.1	RNA polymerase sigma factor	rpoS	K	−1.15	−0.19	0.12
WIV41139.1	Heavy metal response regulator transcription factor	copR	KT	−1.29	−0.81	−0.69
WIV38112.1	Heme ABC transporter ATP-binding protein	modC	P	0.91	0.89	−13.29
WIV39570.1	Co^2+^/Mg^2+^ efflux protein	corD	P	−0.88	−0.58	−1.01
WIV39699.1	Fe(3+)-pyochelin receptor	fptA	P	−0.33	−0.75	−1.02
WIV39983.1	Copper-translocating P-type ATPase	cueA	P	−0.60	−0.85	−1.15
WIV40152.1	Mercury resistance system periplasmic binding protein	merP	P	1.08	1.25	0.90
WIV40151.1	Mercury(II) reductase	merA	H	2.27	1.32	0.37
WIV39465.1	TOBE domain-containing protein	modE	H	−0.05	5.27	2.84
WIV38401.1	Lipid A export permease/ATP-binding protein	vcaM	V	1.06	0.64	−0.03
WIV39298.1	NirD/YgiW/YdeI family stress tolerance protein	-	S	0.78	0.92	1.44
WIV41510.1	Multidrug efflux system transcriptional regulator	mexT	K	0.67	0.69	1.02
WIV37667.1	Multidrug efflux RND transporter periplasmic adaptor subunit	mexV	M	0.19	2.20	−0.20
WIV41011.1	TetR/AcrR family transcriptional regulator	comR	K	0.23	−1.64	−0.57
WIV38749.1	ATP-dependent DNA helicase	recG	L	−1.29	−1.66	−0.67
WIV40154.1	Mercury resistance transcriptional regulator	merR	K	1.15	0.83	1.07
WIV37674.1	Response regulator transcription factor	cpxR	K	−2.36	−1.78	−1.45

**Table 7 nanomaterials-15-01404-t007:** Differentially expressed proteins associated with heavy metal resistance in *P. mirabilis* ATCC 7002.

Protein ID	Protein Name	Gene Name	COG	Fold Ratio		
				10 min	30 min	60 min
KGA90072.1	Transcriptional regulator	modE	K	−0.96	−1.38	−0.76
KGA90513.1	Tellurite resistance protein	tehB	HP	0.65	0.85	3.48
KGA91504.1	Phosphate regulon transcriptional regulatory protein	pcoR	K	0.51	0.02	2.61
KGA92336.1	Protein YgiW	ygiW	S	0.41	0.85	2.13
KGA92342.1	Sensor protein	cpxA	T	−0.75	−0.39	3.75
KGA89141.1	Acetate operon repressor	iclR	K	1.09	0.17	0.85
KGA89884.1	Hypothetical protein DR94_1398	fetA	S	−1.03	−0.37	−0.61
KGA91744.1	Blue copper oxidase	cuiD	Q	0.46	2.46	3.58
KGA92032.1	Disulfide interchange protein	dsbA	O	0.23	1.21	2.09
KGA88977.1	terD domain protein	terZ	T	−2.75	−1.56	1.20
KGA91967.1	Low-affinity inorganic phosphate transporter 1	pitA	P	−1.10	−0.51	−2.31
KGA89008.1	Tellurite resistance protein	terB	P	0.94	1.60	0.04
KGA89027.1	Tellurite resistance protein	terD	T	0.49	1.29	0.61
KGA89035.1	Outer membrane protein	tolC	S	0.62	1.24	0.13
KGA89981.1	Molybdate ABC transporter, periplasmic molybdate-binding protein	modA	P	0.34	1.18	0.99
KGA90832.1	Methionine gamma-lyase	mdeA	E	0.99	1.96	−0.54
KGA91860.1	Phosphate ABC transporter, ATP-binding protein	pstB	P	0.21	1.12	−0.27
KGA91879.1	Phosphate ABC transporter, phosphate-binding protein	pstS	P	1.00	1.74	0.49
KGA92341.1	Transcriptional regulatory protein	cpxR	K	0.38	1.03	0.07
KGA91352.1	Glutamate decarboxylase	gadA	E	−1.04	−1.21	−0.92
KGA89707.1	Copper homeostasis protein	cutC	P	−3.04	−3.45	−1.68
KGA89877.1	ABC transporter family protein	yfeB	V	−1.74	−1.71	−1.08
KGA89026.1	LPS-assembly protein	lptD	M	1.40	1.55	1.18
KGA91809.1	Cadmium-translocating P-type ATPase	zntA	P	1.51	2.60	2.35
KGA92136.1	Metal-binding protein	zinT	S	1.06	1.06	1.78

## Data Availability

Authors can provide raw data upon request.

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
