# Peer review of "Species-Specific Stress Responses to Selenium Nanoparticles in *Pseudomonas aeruginosa* and *Proteus mirabilis"

_nanomaterials, 2025, doi:10.3390/nano15181404_

Round 1

Reviewer 1 Report

Comments and Suggestions for Authors

The present manuscript contains results that are relevant to the scientific community. The following revisions are recommended:
-The originality of the present study in relation to earlier research examining the antibacterial properties of SeNPs should be emphasised in the introduction.
- How was the SeNP concentration (32 ppm) chosen for treatment? In order to elucidate the dose-response effect and facilitate a more robust comparison with other studies, it is recommended that a test with different concentrations, incorporating MIC determination, be conducted.
- It is imperative that negative and positive controls are incorporated into the methodology and results of the antibacterial activity test.
- It is recommended that the overall discussion be improved by means of a more thorough exploration of the results obtained, with the inclusion of additional information from previous studies on SeNPs.

Reviewer 2 Report

Comments and Suggestions for Authors

The research included in the manuscript concerns the effect of selenium nanoparticles on uropathogenic microorganisms. Urinary tract infections are the most common bacterial infections in humans and their treatment is often difficult, especially in times of increasing drug resistance of bacteria. For this reason, the presented research concerns current topics and I consider it important. However, I have a few comments regarding the presented results and the preparation of the manuscript, which I include below point by point.

- why the title of the article includes only Proteus mirabilis and Pseudomonas aeruginosa even though a wider panel of uropathogenic bacteria was studied ?

- the description of the method for testing the antibacterial activity of nanoparticles does not specify how many times the experiment was repeated and what constituted the experimental controls

- What was the reasoning behind choosing this concentration of nanoparticles (32 ppm) and why was only one concentration tested? It's difficult to draw conclusions from this that any microorganism is sensitive or not tested for antibacterial activity.

- in the remaining methods the authors provide "After Se NP treatment,...) however the conditions of action of the anoparticles are not given. Were they the same as in subchapter 2.1 or were all times included?

- In Figure 1, the bacterial samples with nanoparticles should be compared with the control without the antibacterial agent

- The authors provide the following "Notably, previous studies have demonstrated that Se NPs exhibit broad-spectrum antibacterial activity against multiple bacterial species, including P. aeruginosa, Salmonella spp., and Streptococcus mutans”" however, there is no information whether these studies yielded different results; the concentrations tested were the same, and there is simply no comparison of their data with previously published data. This is especially true given that the cited studies also concerned Pseudomonas aerugionsa.

- The description of Figure 2 does not specify how the bacteria were treated with nanoparticles. Furthermore, these images show completely destroyed cells, making it difficult to conclude the mechanism of action of the nanoparticles. Perhaps if the images had been taken after a shorter exposure time, the first stages of the nanoparticles' action would have been better captured.

Reviewer 3 Report

Comments and Suggestions for Authors

The authors have done a commendable work in writing this manuscript to report the stress reponse caused by selenium nanoparticles in Pseudomonas aeruginosa and Proteus mirabilis. However there are a few points that need to be addressed before considering the manuscript further. 

  1. The introduction section can have a table where the use of nanoparticles/nanotechnology enabled therapeutic strategies in UTI treatment with some latest references. 
  2. The nanoparticle synthesis by LASER ablation seems very interesting. Could the authors add a video of the process in the supplementary files?
  3. The photographs of the bacterial strain's inoculation and studies which were done can also be added in the supplementary information. 
  4. Appreciations for the SEM study of the microbes. Could you detail more on the sample preparation needed for this. 
  5. In depth explanation is needed for the nanoparticles' interaction at the cellular level. 
  6. Were the COG functional analysis and KEGG pathway analysis done in triplets ?
  7. Has the python script used for cluster protein expression  made available for reading?

Round 2

Reviewer 1 Report

Comments and Suggestions for Authors

The authors responded to all comments and the manuscript was improved.